# Recent Advance of Liposome Nanoparticles for Nucleic Acid Therapy

**DOI:** 10.3390/pharmaceutics15010178

**Published:** 2023-01-04

**Authors:** Yongguang Gao, Xinhua Liu, Na Chen, Xiaochun Yang, Fang Tang

**Affiliations:** 1Tangshan Key Laboratory of Green Speciality Chemicals, Department of Chemistry, Tangshan Normal University, Tangshan 063000, China; 2The Institute of Flexible Electronics (IFE, Future Technologies), Xiamen University, Xiamen, 361005, China

**Keywords:** gene therapy, nucleic acid, liposome, nanoparticle, chemical molecule

## Abstract

Gene therapy, as an emerging therapeutic approach, has shown remarkable advantages in the treatment of some major diseases. With the deepening of genomics research, people have gradually realized that the emergence and development of many diseases are related to genetic abnormalities. Therefore, nucleic acid drugs are gradually becoming a new boon in the treatment of diseases (especially tumors and genetic diseases). It is conservatively estimated that the global market of nucleic acid drugs will exceed $20 billion by 2025. They are simple in design, mature in synthesis, and have good biocompatibility. However, the shortcomings of nucleic acid, such as poor stability, low bioavailability, and poor targeting, greatly limit the clinical application of nucleic acid. Liposome nanoparticles can wrap nucleic acid drugs in internal cavities, increase the stability of nucleic acid and prolong blood circulation time, thus improving the transfection efficiency. This review focuses on the recent advances and potential applications of liposome nanoparticles modified with nucleic acid drugs (DNA, RNA, and ASO) and different chemical molecules (peptides, polymers, dendrimers, fluorescent molecules, magnetic nanoparticles, and receptor targeting molecules). The ability of liposome nanoparticles to deliver nucleic acid drugs is also discussed in detail. We hope that this review will help researchers design safer and more efficient liposome nanoparticles, and accelerate the application of nucleic acid drugs in gene therapy.

## 1. Introduction

As new gene therapy drugs, nucleic acid molecules have become a research hotspot in the international frontier field, showing outstanding advantages in the treatment of cardiovascular diseases, immune system diseases, nervous system diseases, metabolic diseases, and tumors [1,2,3,4]. The concept of “gene therapy” was first proposed in 1972 [5]. Martin Cline, a blood geneticist, first introduced recombinant DNA into two patients with thalassemia in 1980 [6]. Although the therapeutic effect was not ideal, this event opened up a precedent for the clinical application of gene therapy. It was in 1990 that the first successful clinical trial of a nucleic acid drug for the treatment of severe combined immunodeficiency disease (SCID) was conducted [7]. Early clinical trials have shown that nucleic acid drugs are effective in treating leukemia, thalassemia, and Parkinson’s disease, and can even restore sight to the blind [8,9,10]. However, due to the lack of targeting of nucleic acid drugs and the existence of an immune response, there have been cases of death due to toxicity in clinical trials, which seriously hindered the development of gene therapy [11,12].

With the rise of gene-editing technology, gene therapy has again attracted people’s attention. In 2013, Kynamro^TM^, the first nucleic acid drug for systemic administration, was approved by the US Food and Drug Administration (FDA), which initiated a new era of gene therapy [13]. In 2019, Zolgensma, a nucleic acid drug used to treat spinal muscular atrophy (SMA), was approved by the FDA, further stirring up research on nucleic acid drugs [14]. In recent years, the novel coronavirus has spread around the world, and nucleic acid drugs have played a huge role in the fight against the novel coronavirus [15]. It is conservatively estimated that the global nucleic acid drug market will exceed $20 billion by 2025 [2]. In nucleic acid drugs, nucleic acid molecules play a role in two ways. One is to start from the source of nucleic acid, directly modify the structure of nucleic acid, and directly add therapeutic genes into the genome or delete pathogenic genes from the genome. The other is to start from the expression products of genes by introducing inhibitors or promoters to prevent the expression of pathogenic genes, or promote the expression of therapeutic genes. The nucleic acid drugs mainly include DNAs and RNAs. In RNA-based nucleic acid drugs, compared with miRNA and mRNA, siRNA is widely used, and the number of siRNA-based nucleic acid drugs approved by the FDA is the largest. This review therefore mainly focuses on the application of siRNA. The design and synthesis of nucleic acid drugs are not difficult. However, the shortcomings of nucleic acids, such as poor biological stability, easy degradation by nuclease in vivo, low bioavailability, and low concentration in target tissues, greatly limit the clinical application of nucleic acid drugs. Therefore, the surface modification of liposome nanoparticles was reviewed in this paper. Different modifications endow them with specific functions, such as targeting of cells or tissues and visualization of nanoparticles. Through the study of the structure–activity relationship, the cytotoxicity of liposome nanoparticles can be reduced and the transfection efficiency can be improved. We hope this review can provide a reference for researchers in the design and synthesis of liposome nanoparticles in the future.

## 2. Nucleic Acid Drugs

Nucleic acids, one of the most basic substances of life, were first extracted from pus cells in 1869 by Miescher, who named them nuclein [16]. The names of nucleic acids were not recognized until 1889. The discovery of nucleic acid drugs not only breaks the traditional idea that nucleic acid can only carry genetic information, but also provides a powerful molecular tool for the development of biomedical and biosensing fields. In recent years, with the deepening of genomics research, researchers have found that the generation and development of many diseases are related to genetic abnormalities. Therefore, nucleic acid drugs are gradually becoming a new boon in the treatment of incurable diseases (especially tumors and genetic diseases) [17]. Theoretically speaking, nucleic acid drugs have great advantages in the treatment of some diseases. The patients can obtain a long-lasting therapeutic effect through a single treatment. In addition, nucleic acid drugs are simple in design, mature in synthesis, and have good biocompatibility. Therefore, they have become a new type of drug after small molecule drugs, protein drugs, and antibody drugs. Nucleic acid drugs mainly include DNAs, RNAs, and antisense oligonucleotides (ASOs). RNA drugs include small interfering RNA (siRNA), microRNA (miRNA), and messenger RNA (mRNA). ASO drugs include antisense DNA, antisense RNA, and ribozyme. Currently, 15 kinds of nucleic acid drugs (Table 1) have been approved for marketing, including 2 kinds of naked plasmid DNAs, 9 kinds of ASOs, and 4 kinds of siRNAs, and more than 50 kinds of nucleic acid drugs are in various stages of clinical trials.

### 2.1. DNA

DNA (deoxyribonucleic acid), as the encoding material and carrier of biological genetic information, has extremely important value and significance for research in the field of biomedicine. Watson and Crick discovered the double-helix structure of DNA and the principle of complementary base pairing in the 1950s [18], which led to a deeper understanding of the physical and chemical properties of DNA. In a double-stranded DNA, two single-stranded DNAs are combined to form a double-helix structure by complementary base pairing, where A and T are connected by two hydrogen bonds, and C and G are connected by three hydrogen bonds. The higher the number of intermolecular hydrogen bonds, the lower the energy of the system. Therefore, the maximization of the number of base pairs between the double strands of DNA is the main reason for the stability of the double-helix structure. Plasmid DNA has become a promising nucleic acid drug because of its stable structure, easy preparation, and its ability to be transcribed and translated into the cytoplasm [19]. However, because DNA can change genetic information, how the safety of species can be ensured is a constant concern [20]. In addition, DNA has a loose structure and a negative charge so that it is difficult to be endocytosed by cells, resulting in DNA drugs not working. Currently, some plasmid DNA vaccines have been put into clinical use, but their transfection efficiency is not ideal [21]. Therefore, improving the delivery efficiency of plasmid DNA and the releasability of DNA in the nucleus are an urgent difficulty to be solved.

### 2.2. RNA

#### 2.2.1. siRNA

In gene therapy, RNA interference (RNAi) is a very subversive technology at present, which has attracted the extensive attention of scientists due to its good therapeutic effect [22]. RNAi refers to the introduction of a double-stranded RNA (dsRNA), consisting of sense RNA and antisense RNA, corresponding to mRNA, into cells, which can cause specific degradation of mRNA, leading to gene silencing. In mammalian cells, RNAi can regulate cells to produce a class of double-stranded RNA composed of 21–23 nucleotides, namely small interfering RNA (siRNA). Therefore, siRNA is a kind of RNA that is recognized and synthesized by the Dicer enzyme based on RNAi technology [23]. SiRNA combines with related proteins in cells to form an active protein RNA-induced silencing complex (RISC), and specifically combines with the corresponding mRNA sequence to cut it into small segments with base pairs between 10 and 11, thus blocking the translation of mRNA and the expression of the protein, so as to achieve the goal of treating diseases. SiRNA is a negatively charged bioactive macromolecule, which does not have the ability to target tissues or cells. Its ability to penetrate the cell membrane is extremely poor, and it is also extremely unstable in the physiological environment. Moreover, the transport process of siRNA drugs in cells directly affects their physiological functions. Therefore, the weak transmembrane ability, poor targeting, easy degradation, and short blood circulation time seriously limit the clinical application of siRNA.

SiRNA-based gene therapy is still in its infancy [24]. At present, there are more than a dozen drugs in the clinical stage, and four siRNA drugs have been approved by the FDA for marketing. In 2018, the FDA approved the first siRNA drug, patisiran (Onpattro), which is an siRNA that acts on the liver and is used to treat multiple neuropathies caused by hereditary transthyretin amyloidosis (hATTR) [25]. Recently, the siRNA drugs lumasiran (Oxlumo) and subcliniran (Leqvio), which are used to treat primary hyperoxaluria and adult hypercholesterolemia, have also been approved by the FDA [26,27]. Lumasiran is a subcutaneously injected RNAi therapy that targets the mRNA of hydroxylate oxidase 1 (HAO1), a gene that encodes glycolate oxidase (GO) in the liver. By silencing the HAO1 gene and reducing the expression of the GO enzyme, lumasiran inhibits and normalizes the production of oxalic acid in the liver, thereby halting the progression of PH1 disease. Inclisiran binds to the mRNA that encodes the PCSK9 protein, reducing mRNA levels through RNA interference and preventing the liver from producing PCSK9, thereby enhancing the liver’s ability to remove low-density lipoprotein cholesterol (LDL-C) from the blood and lowering LDL-C levels. Cationic liposomes are the most commonly used delivery carriers for siRNA delivery. They can deliver siRNA into the endosomes, where the concentration of intrinsic immune receptors is very high. This means that the use of cationic lipids to deliver siRNA is particularly vulnerable to immune stimulation. Therefore, chemical modification of the structure of cationic liposomes to reduce the immune response of siRNA to the body is the future research focus of siRNA therapy.

#### 2.2.2. miRNA

MiRNA is the most-studied non-coding ribonucleic acid, consisting of 18–22 nucleotides [28]. Up to now, more than 5000 mRNAs have been identified, which play an important role in a variety of biological and pathological processes [29]. Just like siRNA, single-stranded miRNA can also regulate mRNA translation, and then regulate cell differentiation, proliferation, and other processes. MiRNA drugs mainly consist of two types: antagonists and mimics [30]. MiRNA antagonists regulate mRNA degradation or protein translation by binding to complementary bases of the 3′ untranslated region (UTR) of endogenous miRNA, thereby inhibiting gene expression at the post-transcriptional level [31]. MiRNA mimics are analogues with similar functions as endogenous miRNAs that are synthesized by chemical synthesis. MiRNA mimics can stimulate the high expression of endogenous mature miRNAs in cells, further enhance the silencing effect of endogenous miRNAs, and reduce the expression of proteins in cells, and thus restore the normal physiological function of cells [32]. At present, some miRNA drugs have entered the clinical trial stage, but no drugs have been approved for clinical use.

#### 2.2.3. mRNA

Messenger RNA (mRNA) is a class of single-stranded ribonucleic acids that are transcribed from a strand of DNA as a template and carry genetic information that directs protein synthesis. It has the advantages of large molecular weight, strong hydrophilicity, high biological activity, and easy industrial production [33]. MRNA has many varieties, has an active metabolism, has a short half-life, and can even be decomposed within a few minutes after synthesis. In addition, mRNA is not inserted into a genome such as DNA and viral vectors to affect genetic information, so there is no risk of gene integration, and the safety is high. However, the single-strand structure of mRNA makes it extremely unstable and easy to degrade, and has strong immunogenicity [34]. These shortcomings are important reasons why the previous research on mRNA is far less than that on DNA [35].

At present, the problems of poor stability and low delivery efficiency of mRNA are mainly solved by modifying mRNA structure and introducing delivery vectors [36,37,38]. Mod RNA technology is a safe, efficient, and controllable method of nucleic acid gene therapy. It is through chemical means that one or more nucleotides in the proto-nucleotide sequence are modified to improve the stability of mRNA and achieve the effect of re-encoding mRNA. The introduction of delivery vectors is the most commonly used method to improve mRNA stability and transfection efficiency. Generally, the delivery vectors include liposome nanoparticles (LNPs), exosomes, polymers, and peptides, among which LNPs are the most widely used [39,40,41,42]. Liposomes can be used to protect against COVID-19 by wrapping mRNA into nanoparticles and delivering them into the body [43]. Therefore, research on the delivery of nucleic acid by liposomes has once again become the focus of scientists.

As of December 2022, there have been over 600 million confirmed cases of COVID-19, and the cumulative death toll has exceeded 6.5 million worldwide [44]. The spread of the novel coronavirus has accelerated the marketing of RNA drugs. In 2020, the FDA authorized an emergency use license for a COVID-19 vaccine developed using mRNA technology [45]. Comirnaty (BNT162b2) is the first vaccine launched based on the mRNA technology route in history, and the first COVID-19 vaccine approved in the world with complete phase 3 clinical data. The next year this vaccine was officially approved by the FDA. Clinical results showed that Comirnaty was 91% effective in resisting COVID-19 infection. In 2021, the total revenue of the top ten vaccine companies in the world reached 135.92 billion dollars, and the contribution of COVID-19 vaccines exceeded 100 billion dollars, of which the sales of mRNA-based COVID-19 vaccines reached more than 58 billion dollars [46]. It is believed that the market share will grow with the marketing of mRNA-based tumor vaccine in the future.

### 2.3. ASO

The concept of antisense oligonucleotide (ASO) was first proposed by Zamecnik et al. in 1978 [47]. It refers to a segment of DNA or RNA that is complementary to the target DNA or RNA base and can specifically bind to it. It includes antisense RNA (asRNA), antisense DNA (asDNA), and ribozyme (Rz). Antisense RNA includes endogenous miRNA and exogenous siRNA. ASOs can act specifically on target genes or mRNAs to regulate gene expression from various aspects of gene replication, transcription, splicing, transport, and translation, so as to achieve the purpose of disease treatment. Compared with other nucleic acid drugs, ASO drugs have the largest number of approved drugs on the market. Up to now, 10 ASO drugs have been approved for marketing worldwide [48]: Formivirsen (1998), Mipomersen (2013), Eteplirsen (2016), Nusinersen (2016), Tegsedi (2018), Volanesorson (2019), Golodirsen (2019), Viltolarsen (2020), and Casimersen (2021). More than 50 ASO drugs are in clinical studies. Fomivirsen is the first ASO drug approved by the FDA for the treatment of cytomegalovirus-infected retinitis (CMV) in HIV patients [49]. Due to the development of highly active antiretroviral therapy (HAART), the number of CMV cases has decreased dramatically. So, Fomivirsen did not sell well. ASO drugs gained renewed attention in 2013 with the release of Mipomersen, a drug used to treat homozygous familial hypercholesterolemia (hoFH) [50].

Compared with traditional drugs, ASO drugs have the following advantages: (1) the interaction between traditional drugs and proteins is the van der Waals force, while ASO and pathogenic genes interact through intermolecular hydrogen bonds. Compared with van der Waals force, intermolecular hydrogen bonds have lower energy and stronger stability, so the drug effect is stronger; (2) traditional drugs mainly aim to inhibit the expression of proteins to treat diseases, while ASO drugs inhibit the expression of upstream mRNA, so the drugs have faster and more lasting effects; (3) the synthesis method of antisense nucleic acid is simple and easy for industrial production. As long as the target genes or mRNA of related diseases are found, ASO drugs can be designed and synthesized in a short time. Therefore, ASO drugs have become a hot topic for scientists around the world.

## 3. Liposome Nanoparticles

Naked nucleic acid is quickly degraded by nucleases during blood circulation after intravenous injection, making it difficult for nucleic acid drugs to reach the diseased site. Recent research focuses on the development of safe and efficient nucleic acid delivery vectors. Nucleic acid delivery vectors include viral vectors and non-viral vectors. Previous studies mainly used viral vectors to deliver nucleic acids, including retrovirus vectors and adenovirus vectors. Although these viral vectors can effectively deliver nucleic acids to cells, their potential immune responses and insertional gene mutations have always been a concern [51,52]. In 1999, a young patient died after injecting an adenovirus vector through the hepatic artery [53]. In 2002, two patients with severe combined immunodeficiency developed leukemia-like symptoms after gene therapy with a retroviral vector [54]. These events prompted people to turn their attention to non-viral vectors with high safety [55]. Non-viral vectors mainly include liposomes, polymers, and metal complexes. Among them, liposome is the most-studied and widely used non-viral vector. Liposomes are enclosed nanoparticles with a bilayer membrane which show excellent performance in nucleic acid and small molecule drug delivery.

Lipids have a structure similar to phospholipid bilayers and are generally composed of three parts (Figure 1): a hydrophilic domain, a hydrophobic domain, and a bridging domain (linker) [56]. Hydrophilic domains usually have one or more positive charges and can interact with negatively charged nucleic acids to form liposome/nucleic acid complexes that protect the nucleic acid from nuclease degradation. Hydrophobic domains are usually composed of steroidal compounds or alkyl chains (saturated or unsaturated) and have a great influence on the efficiency of nucleic acid delivery. Some rigid molecules such as benzene rings and naphthalimide can also be used as hydrophobic groups to improve the transfection efficiency [57]. The bridging domains usually take glycerol as the skeleton and connect the hydrophilic and hydrophobic domains with amide, ester, or ether bonds. In general, the ester bonds are able to be biodegraded during systemic circulation. Therefore, the ester bond-linked liposomes have higher release efficiency and less cytotoxicity.

Generally, the stability, membrane fusion, and transfection efficiency of lipids alone after interaction with DNA are very poor, so neutral auxiliary lipids are often needed [58]. Cationic liposomes formed by mixing cationic lipids and neutral auxiliary lipids at a certain molar ratio can greatly improve transfection efficiency. The neutral lipids include dioleylphosphatidylethanolamine (DOPE), 1,2-dioleoglycerol-3-phosphatidylcholine (DOPC), dipalmitoyl phosphatidylcholine (DPPC), and cholesterol (Chol) (Figure 2), of which DOPE is the most commonly used. DOPE may have the characteristics of membrane fusion, which can destroy the membrane of the endosome, promote the escape of the endosome, and promote the entrance of the DNA complex into the cytoplasm, thus enhancing the transfection activity [59].

Cationic liposomes are one of the earliest and most widely used non-viral gene vectors. Liposomes can interact with DNA or RNA to form stable lipoplexes, which can effectively avoid nuclease degradation and increase blood circulation time. In 1987, Felgner et al. synthesized cationic lipid DOTMA containing a glycerol skeleton structure [60]. Two years later, gene transfection mediated with DOTMA was realized in mice for the first time [61]. Since then, many cationic liposomes have been synthesized and have carried nucleic acid into the nucleus, showing high transfection efficiency both in vivo and in vitro [62]. After more than 40 years of development, many highly efficient cationic liposomes have become commercial transfection reagents, some of which have been applied to clinical trials of gene therapy for the treatment of cancer and other familial genetic diseases [63,64,65]. In order to improve the transfection efficiency of liposomes, researchers continuously chemically modify the structure of liposomes, including peptide-modified liposomes, polymer-modified liposomes, dendrimer-modified liposomes, and functional molecule-modified liposomes (Figure 3). After structural modification, the delivery efficiency of liposomes to nucleic acid is further improved, the immune response is reduced, and the targeting towards cells and tissues is increased [66].

### 3.1. Peptide-Modified Liposome Nanoparticles

Peptide-modified liposomes are widely studied as non-viral gene vectors. In general, cationic peptides are used as hydrophilic groups to interact with nucleic acids to concentrate nucleic acids into nanoparticles, so as to achieve the purpose of carrying nucleic acids. These peptide compounds have a small molecular weight and simple structure, can be prepared by solid-phase synthesis, and are easily industrialized. They generally contain multiple amino or guanidine groups. Under the condition of physiological pH value, the amino or guanidine group at the *N*-terminal is in the protonated state, carrying positive charges, so it can have an electrostatic interaction with the negatively charged cell membrane, causing the lipid bilayer structure of the cell membrane to overturn, thus translocating into the cell [67]. These processes can occur even in the absence of receptors and energy intake [68].

Cell penetrating peptides (CPPs) are a kind of small molecular peptide that are most widely used in gene vectors. They are composed of 5–40 amino acids and have the unique function of promoting cell transmembrane transport. At present, more than 100 kinds of polypeptides, including TAT (YGRKKRRQRRR), RGD (Arg–Gly–Asp), Pardaxin (GFFALIPKISSPLFKTLLSAVGSALSSSG GQE), and RVG (YTIWMPENPRPGTPCDIFTNSRGKRASNG), with cell membrane penetration have been found, which can safely and efficiently transfect small molecule drugs, antibodies, polypeptides, nucleic acids, and other substances. Arginine, lysine, and histidine are often used as membrane-penetrating peptides for nucleic acid delivery because of their abundant positive charge and their ability to bind to negatively charged glycoproteins on cell membranes, and play a penetrating role in cell membranes. CPPs can effectively compress nucleic acid molecules through electrostatic interaction to improve the carrier’s inclusion capacity [69]. Moreover, due to their special amino acid sequence and special analysis and purification methods, they have good biocompatibility and biodegradability, which can greatly improve the transfection efficiency of the vector [70].

#### 3.1.1. TAT-Modified Liposome Nanoparticles

TAT is the first polypeptide found to have the function of cell membrane penetration and nuclear localization. It contains the basic domain of the HIV-1 transcriptional activating protein, which can promote the fusion of the HIV virus and cells. Bi et al. prepared cationic liposome TAT-PEG-SN38 modified with TAT as a CPP and PEG as a protection domain and applied it to the study of survivin siRNA delivery [71,72]. The dynamic light scattering experiment showed that the size of liposome nanoparticles ranged from 105 nm to 148 nm, and the *zeta* potential ranged from +3.74 mV to +13.83 mV. TAT-PEG-SN38, after targeted modification by Transferrin (Tf), can significantly increase the toxicity of liposomes and enhance the killing capacity of tumor cells. Western blot results showed that the efficiency of Tf-L-SN38/P/siRNA in silencing the surviving gene was significantly higher than that of TAT-PEG-SN38/siRNA without Tf modification. In addition, Tf-L-SN38/P/siRNA has significant anti-tumor activity. Compared with the control group, the Tf-L-SN38/P/siRNA group inhibited 76.8% tumor volume growth. The mechanism study showed that Tf-L-SN38/P/siRNA entered cells mainly through the Tf- mediated endocytosis pathway, and then internalized through the grid protein-mediated pathway. Tf only partially affected the cell uptake of Tf-L-SN38/P/siRNA.

#### 3.1.2. RGD-Modified Liposome Nanoparticles

RGD is a protein fragment containing an arginine–glycine–aspartic acid sequence. It is a recognition site for the interaction between integrin and ligand proteins, and mediates the adhesion between cells and the extracellular matrix, and between cells [73]. It also has the function of signal transmission and participates in many important life activities [74]. RGD widely exists in various extracellular matrix proteins such as fibronectin, fibronectin, and bone salivary gland protein, and can specifically recognize integrin *α_v_β_3_* and other integrins [75]. The integrin *α_v_β_3_* is highly expressed in a variety of tumor cells and tumor neovascular endothelial cells, and is involved in regulating tumor cell proliferation, angiogenesis, cell migration, and other activities. It is an ideal target for tumor diagnosis and treatment. Bao et al. prepared iRGD-modified liposome nanoparticles iRGD-PEG2000-DSPE, and evaluated its DNA delivery efficiency and therapeutic effect on colorectal cancer mice [76]. Liposome iRGD-PEG2000-DSPE is a hollow sphere with a diameter of approximately 80 nm, and its diameter increases to about 180 nm after wrapping DNA. After the interaction of iRGD-PEG2000-DSPE with DNA, the *zeta* potential decreased from 35–38 mV to 18–22 mV. Western blot analysis showed that liposomes modified with iRGD could efficiently deliver PEDF-DNA into colon cancer cells and tumor sites of mice, and enhance PEDF protein expression. Fluorescence imaging experiments showed that liposome complexes were mainly distributed in the lungs and liver of mice. In addition, the study on the anti-tumor activity of the liposome complex showed that intravenous injection of the liposome complex could effectively reduce the number of metastatic tumor nodules in the lung and prolong the survival time of lung metastatic model mice inoculated with CT26 colorectal cancer cells, indicating that iRGD-PEG2000-DSPE nanoparticles could be used as a potential delivery carrier to treat metastatic colorectal cancer [77]. Ren et al. prepared RGD and doxorubicin hydrochloride (DOX) modified cationic liposome RGD-LP/DOX spherical nanoparticles with a diameter of approximately 131 nm [78]. The uptake of RGD-LP/DOX by HRT-18 colorectal cancer cells was higher than that of liposome LP without RGD modification. RGD-LP/DOX nanoparticles enter cells mainly through macropinosis. In vivo imaging studies further showed that RGD-LP was mainly enriched in tumor tissues of mice and had a significant inhibitory effect on tumor growth.

The targeting effect of RGD is affected by many factors, such as the stereo conformation of RGD, the interaction between cells, and the interaction between ligands and receptors. Khabazian et al. investigated the effect of cationic liposome nanoparticles modified with cyclic RGD on siRNA delivery [79]. After modification with cyclic RGD, liposomes can efficiently deliver *anti*-STAT3 siRNA into melanoma cells and show a good level of internalization and high cytotoxicity. The ability to induce apoptosis of melanoma cells was more than twice that of unmodified liposomes. In vivo imaging experiments showed that RGD-modified liposomes and siRNA complexes were mainly enriched at tumor sites in B16F10 tumor-transplanted mice. However, this liposome is composed of a variety of compounds, including DSPC, PEG2000, DOTAP, and cholesterol. Its complex composition makes it difficult to be applied in clinical research. Although cationic peptides have high encapsulation efficiency and cell uptake, their transfection efficiency is low due to the poor effect of endosome escape and lysosomal degradation. Therefore, enhancing the escape ability from endosomes and escaping lysosomal degradation are main problem to be solved by peptide liposome nanoparticles in the future [80,81].

#### 3.1.3. Pardaxin-Modified Liposome Nanoparticles

Pardaxin is a polypeptide with an *α*-helix structure, composed of 33 amino acids (GFFALIPKIISSPLFKTLSAVGSALSSSGGQE), which obtained its name because it was first found in the pardachirus marmoratus. It can induce apoptosis, inhibit cell proliferation, and promote cell differentiation [82]. Pardaxin can quickly insert into the phospholipid bilayer of the cell membrane, destroy the structure of the phospholipid bilayer, and create gaps in the cell membrane, thus changing the permeability of cells and promoting the ability of cells to absorb external substances [83,84]. In addition, pardaxin has a targeting effect on the endoplasmic reticulum. The cationic liposomes modified with pardaxin are mainly located in the endoplasmic reticulum through the intracellular transport of the non-lysosomal pathway, which can not only deliver bioactive small molecules but also efficiently deliver nucleic acid molecules [85,86]. Compared with liposomes without pardaxin modification, pardaxin-modified liposomes can greatly improve the transfection efficiency of liposomes. Qin et al. constructed pardaxin-modified cationic liposome nanoparticles PAR-Lipo, and studied their physicochemical properties, stability, and safety [87]. Compared with commercial transfection reagent Lipofectamine™2000, PAR-Lipo mediated P53 and PTEN transfection and has higher gene delivery efficiency in vitro and a better anti-tumor effect in vivo. The study on transport mechanisms shows that PAR-Lipo enters the cell mainly through the endocytosis pathway mediated by caveolin, and moves to the endoplasmic reticulum (ER) along the cytoskeleton, effectively avoiding the acidification and degradation of lysosomes, and then enters the nucleus through ER and NE membrane exchange [88]. This special endocytosis mechanism and intracellular transport pathway are the main reasons for enhanced gene transfection efficiency.

#### 3.1.4. RVG-Modified Liposome Nanoparticles

Rabies virus glycoprotein (RVG) is a highly efficient CPP that has good targeting to brain tissue. Because of its advantages such as neurophilicity, blood–brain barrier permeability, and biological safety, it is widely used in the development of brain-targeted drug carriers. However, the loose protein structure and low drug-loading efficiency seriously affect the therapeutic effect on brain diseases. Liposome nanoparticles have regular morphology and size suitable for cell endocytosis. RVG combined with liposome nanoparticles can rapidly deliver nucleic acid drugs into the brain, providing an effective treatment method for the diagnosis and treatment of brain diseases. Rodrigues et al. prepared liposomes modified with RVG and transferrin, which formed spherical nanoparticles with DNA, and could penetrate the blood–brain barrier to deliver plasmid DNA into primary neuron cells without being degraded by nuclease [89]. In the transfection of neuron cells, the transfection effect of liposomes modified with RVG transferrin was significantly better than that of the single protein or unmodified liposomes. Mechanism studies showed that liposomes/DNA were internalized by cells in an energy-independent manner. Gold liposomes can form 20–50 nm nanoparticles after being modified with RVG, which is much smaller than unmodified liposomes and easier to be internalized by cells. It can deliver oligonucleotide miRNA inhibitors into U87 cells to inhibit the expression of miRNA-92b, thereby inhibiting the growth of malignant glioma [90]. In addition, liposomes can cross the blood–brain barrier to deliver ApoE2 encoding plasmid DNA (pApoE2) into brain tissue after double modification with cell-penetrating RVG peptide and penetratin (Pen), which has a good therapeutic effect on Alzheimer’s disease.

### 3.2. Polymer-Modified Liposome Nanoparticles

#### 3.2.1. PEI-Modified Liposome Nanoparticles

Polyethyleneimine (PEI) is a commercial transfection reagent which is called the “gold standard” of transfection reagents [91]. It has a high positive charge density and can effectively compress nucleic acid to form stable polyplexes through charge attraction. The amino group in PEI can combine with the hydrogen ion in the lysosome to form a “proton sponge effect”, which increases the pH in the lysosome, leading to the change in protein structure in the lysosome, thereby inhibiting the activity of the degrading enzyme, so as to achieve the efficient expression of transfected genes [92]. However, in the presence of serum, the particle size of the complex formed by PEI and nucleic acid is large, making it difficult to be endocytosed by cells, and the polyplexes easily combine with the negative components in the blood and are cleared by the reticuloendothelial system, thus reducing the transfection efficiency.

Because cationic liposomes can bind to plasma proteins and aggregate with each other, the PEI/DNA complex encapsulated with cationic liposomes can effectively avoid the clearance of nucleic acid drugs by the endothelial system. Pinnapireddy et al. prepared DOPE/DPPC/Cholesterol (DDC) liposome nanoparticles modified with linear PEI and branched PEI [93]. Liposomes can encapsulate PEI into spherical nanoparticles less than 220 nm. Compared with branched PEI, the DDC nanoparticles modified with linear PEI are more regular in shape, more consistent in size, and less prone to aggregation, which is crucial for the formation of nucleic acid/liposome complexes. The PEI recovery experiment further confirmed the above results. After the combination of liposomes with linear PEI and branched PEI, the recovery of free PEI was 9% and 15%, respectively. Compared with simple PEI vectors, PEI-modified liposome nanoparticles have lower cytotoxicity and higher transfection efficiency, among which linear PEI-modified liposomes are more prominent, mainly because the shielding effect of liposomes reduces the positive charge on the surface of PEI, and the shielding effect on linear PEI is stronger. Recently, Sun et al. prepared the bifunctional liposome nanoparticles LP-PEI1800-SPION modified with PEI and ferric oxide, with the size of the nanoparticles ranging from 130 nm to 250 nm [94]. When the weight ratio of LP-PEI1800-SPION to siRNA was 20, the transfection efficiency was the highest. In vitro MRI imaging experiments show that LP-PEI1800-SPION can deliver siRNA to HepG2 and SMMC-7721, and has a good cell imaging ability, but the imaging effect in vivo needs further investigation.

#### 3.2.2. Chitosan-Modified Liposome Nanoparticles

Chitosan (*β*-(1,4)-2-amino-2-deoxy-*D*-glucan) is a deacetylated chitin, which is a natural, non-toxic, biodegradable, and low immunogenic cationic polymer. Chitosan contains both amino and hydroxyl groups in its molecule, which can be modified with esterification/amidation and halogenation to obtain various chitosan derivatives with different structures and excellent performance. Therefore, chitosan is widely used in pharmaceutical research, the pharmaceutical industry, and biomaterial engineering [95,96]. In recent years, with the rapid development of gene therapy, the study of nucleic acid delivery by chitosan polymers has attracted extensive attention [97].

In 1995, Mumper et al. prepared chitosan and a DNA complex solution [98]. Dynamic light scattering showed that the diameter of chitosan/DNA particles was 150–450 nm, which laid a foundation for the application of chitosan in gene therapy. Due to the existence of a large number of intramolecular hydrogen bonds in chitosan, it is difficult for chitosan to dissolve in water and common organic solvents, so it is easily engulfed by the reticuloendothelial system during the delivery of nucleic acid drugs, reducing the transfection efficiency. The introduction of carboxyl or other hydrophilic groups into chitosan molecules through condensation reactions can significantly improve the solubility of chitosan and greatly improve the transfection efficiency of the vectors. In recent years, a series of hydrophilic water-soluble chitosan gene carriers have been developed, such as carboxyethyl, quaternary ammonium salt, and succinic acid-modified chitosan, which have shown superior performance in the treatment of cancer, leukemia, and genetic diseases [99,100,101,102]. Baghdan et al. prepared low-molecular-weight chitosan nanoparticles LCPs encapsulated in DPPC/Cholesterol liposomes and used them in the study of plasmid DNA delivery [103]. The liposomes modified with chitosan have lower cytotoxicity and better DNA protection, and the transfection efficiency can be more than twice that of unmodified chitosan. The study of chicken embryo chorioallantoic membrane model in vivo shows that LCPs can efficiently infect the chorioallantoic membrane (CAM) without damaging the surrounding blood vessels, showing good biocompatibility. Fihurka et al. prepared bifunctional liposome nanoparticles by encapsulating the chitosan/siRNA complex with cannabinoid-modified lipid, and applied them to the treatment of Huntington’s disease model mice [104]. The experimental results showed that YAC128 transgenic mesenchymal stem cells, after being transfected with siRNA coated nanoparticles, showed the dual effects of reducing the expression of mutant HTT gene and increasing the expression of inflammatory cytokine IL-6. A carrier simultaneously carries nucleic acid drugs and chemical drugs, and accurately delivers the two drugs to the lesion site, which can greatly enhance the therapeutic effect of drugs. Increasing the targeting of the carrier and reducing the side effects of drugs are the developmental direction of the carrier in the future.

#### 3.2.3. PEG-Modified Liposome Nanoparticles

Polyethylene glycol (PEG) is a kind of linear polymer macromolecule with high hydrophilicity which is obtained from ethylene oxide and water through additive polymerization, or from ethylene glycol monomer through gradual additive polymerization. PEG has strong hydrophilicity, good biocompatibility, non-toxicity, no side effects, and no immunogenicity, so it is often used to modify different biomaterials [105]. After PEG modification, the liposome nanoparticles form a layer of hydration membrane on the surface, covering the hydrophobic binding sites, reducing the van der Waals force of plasma protein and liposomes, thus shielding the recognition and uptake of liposomes by the reticuloendothelial system (RES) and prolonging the cycle time in vivo [106]. Therefore, such liposomes are also called stealth liposomes. Yoshiyuki et al. prepared three PEG-modified quaternary ammonium cationic liposomes PEG-DSG, PEG-Chol, and PEG-CS, and evaluated their siRNA delivery ability [107]. The dynamic light scattering experiment showed that the diameter of the liposome nanoparticles was 90–120 nm, and *zeta*-potential was 37–53 mV. The PEG-liposome/siRNA complex can greatly reduce the efficiency of gene silencing in cells. In vivo experiments showed that intravenous injection of the liposome/siRNA complex could significantly reduce the accumulation of siRNA in the lungs and prolong the circulation time of siRNA in the body.

PEG can increase the stability of liposomes, escape the phagocytosis of macrophages, and effectively prolong the blood circulation time of drugs in the body [108]. With the increase in PEG molecular weight, the recognition and uptake of liposomes by the shielding reticuloendothelial system (RES) are gradually enhanced. Bedu et al. studied the effect of PEG with molecular weights of 1–3 kDa, 5 kDa, and 12 kDa on the stability of lipid DPPE [109]. The results showed that PEG (1–3 kDa) and PEG (5 kDa) were easy to fuse with DPPE to form micelles, while 12 kDa PEG was difficult to fuse with DPPE, resulting in two-phase separations. These results indicate that short-chain PEG can form a miscible lipid bilayer with cationic lipids, which is more suitable for drug-carrier modification. The stability of the liposome polymer is also closely related to the content of PEG. The higher the concentration of PEG, the more effective the steric hindrance of PEG should be, and the better the inhibition effect on the aggregation of liposomes [110].

Although the steric hindrance of PEG increases the stability of liposomes, it also inhibits the uptake of liposomes by target cells, preventing the “escape of endosomes” in pH sensitive liposome cells [111]. In order to solve the above problems, researchers began to try to introduce cell-targeted small molecule compounds to increase the intake of liposomes by target cells. Tang et al. introduced folic acid molecules into 1,2-distanoyl-*sn*-glycero-3-phosphoethanolamine (DSPE) FA-PEG5000-DSPE modified with PEG of different molecular weights (MW: 2.0 kDa, 3.4 kDa, and 5.0 kDa), and studied their delivery effects on siRNA [112]. The experimental results showed that polo-like kinase 1 (PLK1) siRNA transfection mediated by FA-PEG5000-DSPE could selectively inhibit the growth of human nasopharyngeal carcinoma KB cells. In vivo experiments showed that the liposome modified with PEG and folic acid at the same time, without affecting the transfection efficiency, could reduce the aggregation of liposomes and red blood cells, and increase the stability of liposomes.

### 3.3. Dendrimer-Modified Liposome Nanoparticles

A dendrimer is a kind of synthetic compound with a dendriform topology. It has controllable molecular weight and size, a highly branched structure, and a unique monodispersity, making it have special properties and functions [113]. It is composed of a central core, an inner layer repeating unit, and an outer layer terminal group. It has a high degree of geometric symmetry, accurate molecular structure, a large number of surface functional groups, and internal cavities. According to the different molecular weights, the particle size of the dendrimer ranges from 1 nm to 13 nm. Dendrimers are mainly synthesized by divergent and convergent methods. In 1985, Tomalia et al. first synthesized “polyamide amine” (PAMAM) dendrimers through divergent methods [114]. At present, more than 100 different families of dendrimers have been synthesized, including polypropyleneimine, polylysine, and poly(benzyl ether). They are widely used in industry, agriculture, medicine, and other fields [115].

PAMAM is a typical dendrimer with the most extensive research and application. It is usually synthesized by the method of alternating the covalent coupling divergence of methyl acrylate and ethylenediamine with ethylenediamine as the center. According to the difference of the outermost connecting molecules, the whole-generation PAMAM with an amino terminal group and the half-generation PAMAM with carboxyl terminal group can be obtained. The PAMAM surface contains a large number of amino groups. With the increase in generation, the number of primary amino groups also doubled. At present, the eighth generation PAMAM with 1024 amino groups has been synthesized [116]. Due to its advantages of high branching, high symmetry, and high controllability, PAMAM has been widely used in drug carrier, biosensing, and NMR imaging.

PAMAM has a unique stars-like three-dimensional structure, good biocompatibility, and a large number of modifiable amino groups, which make it show superior performance in the process of gene delivery. The amino group in PAMAM is easy to protonate under acidic conditions. It can play the role of a “proton sponge” in acidic organelles and has a strong proton buffer capacity, which can help nucleic acid complexes escape from the endosome/lysosome and effectively avoid the degradation of nucleic acid. In general, the more amino groups in PAMAM, the stronger the “proton sponge” effect. However, excessive amino groups cause hemolysis and cytotoxicity. At present, the toxicity of gene carriers is mainly reduced by modifying the surface amino groups, such as liposomes, amino acids, and folic acid. Feng et al. synthesized the first generation PAMAM-modified cationic liposome, in which lauric acid was used as hydrophobic moiety, amide bond was used as linking moiety, and lysine-modified PAMAM of generation 0 was used as hydrophilic moiety [117]. The particle size of the complex formed by liposome and DNA gradually decreases with the increase in liposomes. When *w*/*w* ≥ 10:1, the particle size is less than 300 nm, and the zeta potential is approximately 10 mV, which can be endocytosed easily by cells. The mechanism study showed that the transfection efficiency of liposome decreased significantly at lower temperatures, which could suggest that the liposome entered the cell through energy-dependent endocytosis. Further studies showed that caveolin-mediated endocytosis was the main way for liposomes to enter cells, and liposomes could effectively avoid lysosomal degradation after entering cells. Imran et al. encapsulated the fifth-generation PAMAM with DPPC liposome nanoparticles to prepare DPPC/CH-PAMAM lipodedriplexes [118]. They showed excellent performance in the plasmid DNA transfection experiment, luciferase gene silencing experiment and cell endocytosis experiment.

The codelivery of small molecules and nucleic acids has always been a challenge for drug carriers. Recently, Hu et al. reported a nano delivery system (siL1@PM/DOX/LPs) with the fourth-generation PAMAM as the center and pH-sensitive liposome as the shell [119]. It has good biocompatibility and excellent antiserum performance, and can simultaneously deliver programmed cell death ligand 1 (PD-L1) siRNA and doxorubicin (DOX) into MCF-7 cells, which has a significant inhibitory effect on tumor cell proliferation and PD-L1 expression. In vivo experiments show that siPD-L1@PM/DOX/LPs are highly enriched in the tumor site and significantly inhibit tumor growth. The codelivery of small molecule drugs and nucleic acid drugs can greatly enhance the therapeutic effect on tumors and reduce the side effects of drugs, which is likely to become an important direction of future liposome research.

### 3.4. Multifunctional Liposome Nanoparticles

The liposome is the earliest and most widely used non-viral nano delivery system. However, the current liposome delivery system still has many defects such as low drug loading, poor targeting, and poor stability of internal circulation. On the one hand, multifunctional liposomes can effectively solve the inherent defects of liposomes and improve their transfection efficiency; on the other hand, they can enable liposomes to play a role other than nucleic acid delivery and expand their application scope. For example, liposomes modified with fluorescent molecules can not only dynamically monitor the delivery process of nucleic acids, but also be used as fluorescent probes to identify metal ions and small biological molecules. According to their different functions in nucleic acid delivery, multifunctional liposomes are mainly divided into three categories: fluorescent liposomes, magnetic imaging liposomes, and receptor-targeted liposomes.

#### 3.4.1. Fluorescent Liposomes

Fluorescence sensing technology realizes the specific recognition of the guest molecule through the changes in fluorescence intensity, fluorescence lifetime, and emission wavelength before and after the interaction between the fluorescent molecule and the guest molecule [120]. There are many detection mechanisms of fluorescent probes, mainly including photoinduced electron transfer (PET), intramolecular charge transfer (ICT), fluorescence resonance energy transfer (FRET), twisted intramolecular charge transfer (TICT), and excited state intramolecular proton transfer (ESIPT). As early as the 1960s, Goppelsröder found that after the interaction of morin dye with Al^3+^, the fluorescence was significantly enhanced [121], realizing the detection of Al^3+^. Nowadays, fluorescence sensing technology based on small molecule fluorescence probes has been widely used in food detection, environmental monitoring, disease diagnosis, and other fields. With the rapid development of microscopic imaging technology, fluorescent probes have shown great potential for in situ non-invasive imaging of organisms.

Fluorescent molecules mainly include naphthalimide, rhodamine B, and fluoroboron fluorescein. Among them, naphthalimides have been widely used in fluorescent dyes, solar cells, biological imaging, and other fields due to their excellent light, thermal and chemical stability, large stokes shift, and easy structure modification. Gao et al. first synthesized [12]aneN_3_-based cationic liposomes modified with naphthalimide [57]. They can compress DNA into spherical nanoparticles of 100–200 nm. With their good fluorescence properties, naphthalimide-modified liposomes can dynamically trace the process of DNA transmembrane, transport, and release. Most importantly, in the luciferase expression experiment, its transfection efficiency was higher than the commercial transfection reagent Lipofectamine 2000. The structure–activity relationship study showed that the transfection efficiency was greatly related to the length of the hydrophobic chain between naphthalimide and [12]aneN_3_ [122]. The longer the carbon chains, the higher the transfection efficiency. The transfection efficiency of liposomes containing 10 carbon atoms in HeLa cells is more than twice that of Lipofectamine 2000. After further modification with cholic acid, the liposomes show a high targeting potential towards liver cancer cells. The study of transport mechanisms showed that the liposome/DNA complex mainly entered cells in an energy-independent manner. Further research found that [12]aneN_3_ compounds modified with naphthalimide can not only be used as gene carriers to deliver nucleic acids, but can also be used as fluorescent probes to identify copper ions and small biological molecules [123,124].

#### 3.4.2. Magnetic Imaging Liposomes

Magnetic resonance imaging (MRI) is one of the most powerful medical diagnostic tools, which has the characteristics of three-dimensional imaging and continuous imaging. It is not only the first choice for imaging the brain and central nervous system, but also the main tool for evaluating the function of heart disease and detecting tumors [125,126,127]. Paramagnetic nanoparticles have the advantages of small size, large specific surface area, good suspension stability, and directional transport and enrichment under the effect of an external magnetic field, which shows great potential applications in the biomedical field [128]. Among many magnetic nanoparticles, Fe_3_O_4_ nanoparticles are magnetic materials with strong magnetism, simple preparation, and good biocompatibility, which are widely used in tumor diagnosis and treatment, immune detection, and gene delivery [129,130].

Because of the small size and large specific surface area of Fe_3_O_4_ nanoparticles, it is easy for them to agglomerate in the process of gene transfer [131]. In addition, ultra-small Fe_3_O_4_ nanoparticles also have the problem of short blood circulation time and difficulty in reaching target organs [132]. Chemical modification of the surface of Fe_3_O_4_ nanoparticles can effectively solve the problems discussed above. Do et al. prepared PEG-encapsulated magnetic nanoparticles in DMAPAP liposomes by reverse-phase evaporation and cosolvent ultrasound [133]. These magnetic cationic liposomes are spherical particles whose mean size is around 200 nm. Under the effect of the external magnetic field, their transfection ability to pFAR4-luc DNA in CT-26 cells was significantly enhanced. Recently, Ye et al. prepared cationic liposome nanoparticles modified with ferric oxide and studied their delivery ability to oxaliplatin (OXA) and MDC1-AS in cervical cancer cells [134]. First, they mixed DPPC, DC-cholesterol, DOAB, and cholesterol in a certain weight ratio, and then added ferric oxide and ammonium sulfate. After hydration, the mixture was extruded into spherical nanoparticles MTCL of about 140.6 nm, which was tested by transmission electron microscope (TEM). The diameter of nanoparticle MTCL-OXA coated with small molecule drug OXA increased to approximately 178.3 nm, and the diameter of nanoparticle MTCL-OXA-MDC1-AS coated with MDC1-AS increased to approximately 350.5 nm. It can be seen that the drug-coated nanoparticles formed by extrusion have a great impact on the particle diameter. In vitro and in vivo experiments show that magnetic thermosensitive liposome nanoparticles MTCL can deliver OXA and MDC1-AS into cervical cancer cells. Compared with OXA or MDC1-AS alone, this synergistic delivery significantly increased the inhibitory effect on cervical cancer cells. However, the complex composition and preparation procedure of liposomes limit their clinical application.

#### 3.4.3. Receptor Targeting Liposome Nanoparticles

Liposome nanoparticles (LPNs) are hollow spherical closed vesicles formed by the self-assembly of a phospholipid bilayer. In the nanoparticle, the hollow part can be loaded with hydrophilic small-molecule drugs and nucleic acids, and the middle of the bilayer can be loaded with lipophilic molecules. Therefore, LPNs can send different drug molecules into cells at the same time for multiple treatments of diseases. LPNs have the advantages of low toxicity, good stability, and simple preparation, so they are widely used in the delivery of chemical drugs and nucleic acid molecules. However, the targeting effect of LPNs on tissues or organs is not ideal, which makes it difficult for drugs to gather in diseased areas, thus causing damage to normal tissues and cells. Improving the targeting of LPNs and reducing the toxic and side effects on normal tissues and organs are the current research focus. Tumor-targeting liposomes are the most widely studied nano-delivery vectors. In order to deliver nucleic acid to specific tumor cells and reduce the toxic and side effects on normal cells, specific ligands can be modified on the surface of liposome nanoparticles to bind with specific receptors on the surface of tumor cells and enter cells through receptor mediated effects. As a tumor-targeting nano-delivery system, the ligands used to modify liposomes include folic acid, galactose, choleric acid, peptide, and nucleic acid aptamer. Among them, folate- and galactose-modified cations have always been the focus of researchers.

Folic acid, also known as pteroylglutamic acid, is a water-soluble vitamin which is named after the abundance of green leaves in plants. Folic acid has the advantages of high stability, weak immunogenicity, and strong binding capacity with its receptor. The folate receptor is almost not expressed in most normal tissues and cells, but are highly expressed in ovarian cancer, colon cancer, cervical cancer and breast cancer [135,136,137]. It is one of the important targets of targeted therapy. Therefore, folate-modified liposome nanoparticles can target the folate receptor on the tumor cell membrane and enter the tumor cells under the folate receptor-mediated endocytosis, so as to achieve the goal of targeted therapy. Lee et al. first synthesized a folate-modified liposome nanoparticle FA-PEG-DSPE and evaluated the delivery effect of doxorubicin (DOX) [138]. The results showed that the intake of FA-PEG-DSPE/DOX by KB cells was 45 times as much as that of PEG-DSPE/DOX without folic acid modification, and the cytotoxicity was 86 times as much. In addition, calcein encapsulated by FA-PEG-DSPE can almost be internalized by HeLa cells with high expression of the folic acid receptor, indicating that folate-modified liposome nanoparticles have good targeting to tumor cells.

Subsequently, folate-modified liposome nanoparticles were widely used in the study of RNA and DNA delivery [139,140]. He et al. synthesized folate-modified cationic liposomes F-PEG-CLPs, which were composed of DOTAP, cholesterol, mPEG-Chol, and F-PEG-suc-Chol in a molar ratio of 50:45:5:0.1 [141]. F-PEG-CLPs can condense plasmid DNA into nanoparticles with a diameter of 193–200 nm and a *zeta* potential of 35–38 mV. They showed high transfection activity in SKOV-3 cells. The addition of free folic acid completely inhibited the uptake of F-PEG-CLPs, indicating that the effect of folate and its receptor enhanced the transfection ability of F-PEG-CLPs. Further studies showed that the folate receptor-targeted liposome could deliver CRISPR plasmid DNA, due to immature delivery technology into SKOV-3 cells, and efficiently express Cas9 and single-strand RNA targeting ovarian cancer-related DNA methyltransferase 1 gene (gDNMT1). In vivo experiments showed that the F-LP/gDNMT1 complex could significantly inhibit the growth of paclitaxel-sensitive and drug-resistant ovarian cancer, and its side effects were less than those of paclitaxel. Using CRISPR-Cas9 technology to edit the genome of cancer cells is expected to become a promising treatment for ovarian cancer [142].

Recent studies have found that folate receptor *β* (FR*β*) is overexpressed in tumor-associated macrophages, and it can mediate folate transport through endocytosis. Tie et al. applied folate-modified liposome F-PLP to the study of BIM-S plasmid delivery [143]. The experimental results showed that transfection of LL/2 cells and MH-S cells mediated by F-PLP could induce apoptosis. Intravenous injection of F-PLP/p-BIM in lung cancer model mice could significantly reduce the number of FR*β* positive cells, and significantly inhibit the growth of tumors and tumor blood vessels. Therefore, the study on the targeting effect of folate-modified liposome nanoparticles on folate receptors FR*β* may provide a new idea for the targeted treatment of lung cancer.

Galactose is an aldose composed of six carbon atoms, which is a component of lactose in mammalian milk. Galactose ligands can bind to the highly expressed asialoglycoprotein receptors (ASGPR) on the surface of liver cells. The conjugated ligand–receptor complexes are endocytosed into lysosomes. After decomposition and metabolism, functional substances with ligands are released, while ASGPR is not degraded and is transported back to the cell membrane to continue to participate in the recognition–endocytosis–efflux cycle [144]. Using this ligand–receptor interaction, the liposome nanoparticles can be modified with galactose to deliver nucleic acid drugs to the liver specifically for the treatment of liver diseases.

Rozema et al. used Dynamic Polyconjugate technology to deliver siRNA into the liver of mice and verified the concept of liver targeting [145]. Sonoke et al. synthesized galactose-modified cationic liposomes, which can concentrate siRNA into 100–140 nm nanoparticles, and transport siRNA into liver cells through sialoprotein receptors that recognize galactose residues [146]. Jiang et al. prepared Gal LipoNP, galactose-conjugated liposome nanoparticles containing siRNA, with a particle size of approximately 116 nm and *zeta* potential of approximately 14 mV [147]. The in vivo administration experiment showed that after intravenous injection of Gal-LipoNP-siRNA-ConA for 6 h in the induced fulminant hepatitis model mice, most of the drugs avoided the degradation of nuclease and accumulated a lot in the liver. Compared with the control group, the levels of ALT and AST in the serum of mice treated with Gal-lipoNP-siRNA decreased significantly, indicating that galactose-modified liposomes have great potential in RNAi treatment of liver diseases. With the spread of the epidemic, the development of mRNA-based COVID-19 vaccine delivery has rapidly become a research hotspot. Guo et al. prepared a new *α*-galactose-modified vaccine delivery vector, *α*-GC-Lip, and successfully translated mRNA into the target protein in the cytoplasm of antigen presenting cells [148]. Further research shows that the *α*-GC-LPR/mRNA complex can significantly increase the expression of bone marrow-derived cells’ (BMDCs) surface molecules and the secretion of cytokines to promote the maturation of dendritic cells, affect the tumor microenvironment, and thus improve the efficiency of tumor immunotherapy.

## 4. Conclusions and Future Directions

Liposome nanoparticles as drug carriers have good biocompatibility, which can improve the stability of drug encapsulation and increase the efficiency of drug uptake by cells. Due to the immaturity of synthesis technology of targeted carriers, nucleic acid drug targets are still mainly limited to liver tissues, and the indications are mostly rare diseases and genetic diseases. Even if it is limited to the stage of liver target at present, the potential of their application is inestimable, especially in the field of chronic diseases such as lowering blood lipid and lowering blood pressure. With the outbreak of COVID-19, liposomes have lived up to expectations and demonstrated their excellent delivery ability for nucleic acid drugs.

Liposome/siRNA drugs have great potential in gene therapy. Once the limitation of liver targeting is broken, its extensive application scenario will bring considerable market space, especially in tumor therapy, along with a huge future demand and development space. MRNA has the advantages of high biological activity and easy industrial production, and it can start protein translation without entering the nucleus, so there is no risk of gene integration. However, the single-strand structure of mRNA makes it extremely unstable and easily degraded. With the gradual maturation of the preparation technology of liposome nanoparticles, the liposome/mRNA vaccine has played a significant role in the COVID-19 epidemic, changing people’s prejudices against mRNA. However, there are still many unknowns about the mechanism of mRNA immunization, and the research on the mechanism is still a long-term focus.

It is helpful to understand the delivery mechanism of nucleic acid drugs by modifying LPNs with fluorescent molecules or magnetic molecules and visualizing the delivery process of nucleic acids. In addition, the introduction of targeted groups in liposome nanoparticles can improve the translation efficiency or expression efficiency of nucleic acid in diseased areas, and reduce the side effects on other tissues and organs. Therefore, the preparation of functionalized liposome nanoparticles will become a key step in nucleic acid therapy. However, due to individual differences, even the same drug delivery technology may have different effects on different human bodies. Therefore, the widespread application of liposome nanoparticles in gene therapy still requires long-term experience accumulation and technical breakthroughs.

## Figures and Tables

**Figure 1 pharmaceutics-15-00178-f001:**
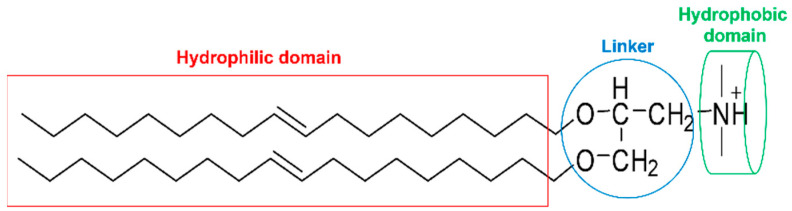
The general chemical structure of lipid.

**Figure 2 pharmaceutics-15-00178-f002:**
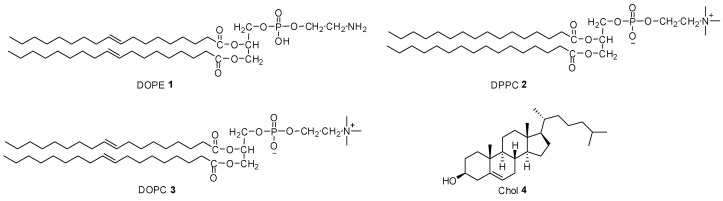
The structure of neutral auxiliary lipids.

**Figure 3 pharmaceutics-15-00178-f003:**
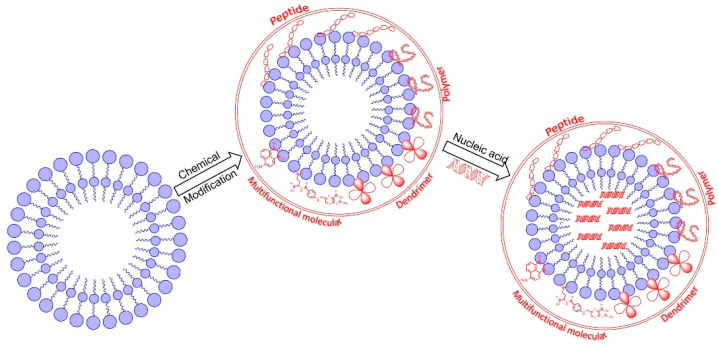
The structure of liposome nanoparticles modified with different chemical molecules.

**Table 1 pharmaceutics-15-00178-t001:** Global nucleic acid drugs approved for marketing.

	Drug Name	Product Name	Company	Drug Target	Indications	Year of Approval
1	Fomivirsen sodium	Vitravene	Ionis Novartis	*CMV UL123*	Cytomegalovirus Retinitis	1998
2	Pegaptanib sodium	Macugen	Valeant	*VEGF-165*	(Wet) Age-related macular degeneration	2004
3	Mipomersen sodium	Kynamro	Ionis Genzyme Kastle	*apo B-100 mRNA*	Homozygous familial hypercholesterolemia	2013
4	Defibrotide sodium	Defitelio	Jazz	*No exact target*	Serious venooclusive disease	2013
5	Eteplirsen	Exondys 51	Sarepta	*DMD exon 51*	Duchenne muscular dystrophy	2016
6	Nusinersen sodium	Spinraza	Ionis and Biogen	*SMN2 exon 7*	Spinal muscular atrophy	2016
7	Inotersen	Tegsedi	Akcea (Ionis)	*TT* *R*	Hereditary transthyretin amyloidosis	2018
8	Patisiran	Onpattro	Alnylam	*TTR-FAP mRNA*	Hereditary transthyretin amyloidosis	2018
9	Golodirsen	Vyondys 53	Sarepta	*DMD exon 53*	Duchenne muscular dystrophy	2019
10	Volanesorsen	Waylivra	Ionis and Akcea	*APOC3*	Familial chylomicronemia syndrome	2019
11	Givosiran	Givlaari	Alnylam	*ALAS1*	Acute hepatic porphyria	2019
12	Lumasiran	Oxlumo	Alnylam	*HAO1 m* *R* *NA*	Primary hyperoxaluria (type I)	2020
13	Inclisiran	Leqvio	Novartis and Alnylam	*PCSK9*	Homozygous familial hypercholesterolemia	2020
14	Viltolarsen	Viltepso	Nippon Shinyaku	*DMD exon 53*	Duchenne muscular dystrophy	2020
15	Casimersen	Amondys 45	Sarepta	*DMD exon 45*	Duchenne muscular dystrophy	2021

## Data Availability

Data sharing not applicable.

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
