# Peer review of "Recent Advance of Liposome Nanoparticles for Nucleic Acid Therapy"

_pharmaceutics, 2023, doi:10.3390/pharmaceutics15010178_

Round 1

Reviewer 1 Report

The current review manuscript provides an incomplete account of recent advances in liposomal delivery of nucleic acids. There are several shortcoming in the manuscript as follows:

1. Very tedious with one 3 figures and no tables. The figures are just schematics.

2. No mention and description of the very important lipoplexes.

3. Big focus on siRNAs but what about micro and mRNAs?

4. Language and composition is a big issues starting with the title itself.

5. What new knowledge is this review bringing except recent advnaces?

Reviewer 2 Report

The review manuscript by Y. Gao. et al., entitled “Recent advance of liposome nanoparticles for nucleic acid therapy” summarized the advances in lipid nanoparticle enabled nucleic acid therapeutic strategies for inherent acquired diseases. Overall, authors summarized the lipid nanoparticle enabled nucleic acid therapeutics well. However, there are few suggestions listed below to further improve the quality of manuscript.  It can be accepted for publication after minor revision.

Major Comments:

1.      Authors should provide the list of lipid nucleic acid therapeutics in clinical trials/FDA approved in tabular form in a disease specific manner.

2.      Authors should provide schematic representation of each type of nucleic acid therapeutics with mechanism for a better understanding.

Minor Comments:

1.      In some places, enthusiasm while reading the manuscript is dampened due to the inappropriate sentences such as in page 5 mentioned that “In general, compared with amide bonds, lipids containing ester bonds have higher nucleic acid release efficiency and higher transfection efficiency” However, it is proven that amide linkers are more stable in in vivo conditions.

2.      In page 5, authors described that “Some rigid molecules such as benzene ring and naphthalimide can also be used as hydrophobic groups to improve the transfection efficiency” authors should prove the proper reference.

3.      A proper reference check is needed.

Reviewer 3 Report

Authors proposed a paper entitled “Recent advance of liposome nanoparticles for nucleic acid therapy” for the publication in Pharmaceutics, MDPI.

 After a careful reading of this review paper, a general comment could be that the scientific soundness is quite high. Moreover, the paper is well written.

I have some revisions to propose to the authors, as follows

1) tumors[1-4]” add a space before reference parenthesis. This simple modification could be provided for all the other cases in the manuscript.

2) Inside the sentence “global nucleic acid drug market will exceed $20 billion by 2025”, this affirmation should require a reference.

3) “Up to now, there have been over 600 million confirmed cases of COVID-19” I suggest adding the date of this sentence, in order not to give immediate information on the period.

 4) Figure 1 is just a representation of a phospholipid. In this image, a focus on the head and on the hydrophobic tails has been proposed.  In my opinion, it is not properly correct to define this sketch as “chemical structure of liposome”. I am sure about the fact that authors know the definition of liposomes, that are generally indicated as spherical vesicles characterized by a double layer of phospholipids and an inner aqueous core. The double layer (or layers) of phospholipids not necessarily have this shape and this kind of head, as correctly the authors say subsequently. Therefore, it is not correct to refer to this image as generic liposome. Therefore, my idea for you is to eliminate this figure, that is also well-known and does not add anything to the topic.

 5) Figure 2 is just a composition of well known phospholipids. I am sorry to say that, also in this case, nothing is added to the study of the literature proposed in this section.

6) This reguards the title of this section: “Chitosan-liposome nanoparticles” are those chitosane-liposome nanoparticles or chitosan-coated liposomes?

7) Also in this case, this reguards the title of this section: “PEG-liposome nanoparticles”, in this case, also, it should be defined as a PEG coating

8) “can effectively solve the above problems” I would say “can effectively solve the problems discussed above”.

9) “are spherical particles less than 200 nm” should be “---spherical particles whose mean size is around 200 nm”.

10) “delivery ability to OXA” be sure that all the acronyms have been defined. Moreover, I suggest adding an abbreviation list, according to the guidelines of this journal.

 11) “MTCL of about 140.6 nm.” This mean size comes from an instrument average or an algebraic average. We are talking about nanometers; therefore, I would approximate this value to unity.

 12) “indicating that the effect of f folate and” there is a “f” that should be eliminated.

 13) “Due to immature delivery technology, nucleic acid drug targets” please expand this concept. Are you referring also to production technologies of drug carriers? Please, clarify.

Round 2

Reviewer 1 Report

The authors did not provide a point-to-point rebuttal to the concerns raised. I checked through the manuscript and the following comments are still pending unless the authors have a rebuttal response for the below:

3. Big focus on siRNAs but what about micro and mRNAs?

5. What new knowledge is this review bringing except recent advances?

Author Response

3. Big focus on siRNAs but what about micro and mRNAs?

Re: The nucleic acid drugs mainly include DNA and RNA. In RNA-based nucleic acid drugs, compared with miRNA and mRNA, siRNA is widely used, and the number of siRNA-based nucleic acid drugs approved by FDA is the largest. This review therefore mainly focuses on the application of siRNA. The above contents are added in the second paragraph of the introduction.

5. What new knowledge is this review bringing except recent advances?

Re: The surface modification of liposome nanoparticles was reviewed in this paper. Different modifications endow them with specific functions, such as targeting of cells or tissues and visualization of nanoparticles. Through the study of structure-activity relationship, the cytotoxicity of liposome nanoparticles can be reduced and the transfection efficiency can be improved. We hope this review can provide a reference for researchers in the design and synthesis of liposome nanoparticles in the future. The above contents are added at the end of the introduction.

Reviewer 3 Report

Paper is now acceptable for pubblication.

Author Response

Thanks